# Withaferin a Attenuates Retinal Ischemia-Reperfusion Injury via Akt-Dependent Inhibition of Oxidative Stress

**DOI:** 10.3390/cells11193113

**Published:** 2022-10-02

**Authors:** Zheyi Yan, Yuanlin Zhang, Chunfang Wang, Yanjie Li, Qiang Su, Jimin Cao, Xiaoming Cao

**Affiliations:** 1Department of Ophthalmology, First Hospital of Shanxi Medical University, Taiyuan 030001, China; 2College of Traditional Chinese Medicine and Food Engineering, Shanxi University of Chinese Medicine, Taiyuan 030619, China; 3Key Laboratory of Cellular Physiology, Ministry of Education, Department of Physiology, Shanxi Medical University, Taiyuan 030001, China; 4Department of Orthopedics, Shanxi Medical University Second Affiliated Hospital, Taiyuan 030000, China

**Keywords:** withaferin A, phytochemical, oxidative stress, apoptosis, endothelium, retinal ischemia-reperfusion injury

## Abstract

**Background:** Retinal ischemia-reperfusion (I/R) injury often results in intractable visual impairments. The survival of retinal capillary endothelial cells is crucial for the treatment of retinal I/R injury. How to protect retinal endothelia from damage is a challenging work. Withaferin A, a small molecule derived from plants, has antibacterial and anti-inflammatory effects and has been used for about millennia in traditional medicine. The present study aimed to investigate the potential protective effect of withaferin A on retinal I/R injury. **Methods:** The drug-likeness of withaferin A was evaluated by the SwissADME web tool. The potential protective effect of withaferin A on the I/R-induced injury of human retinal microvascular endothelial cells (HRMECs) was investigated using multiple approaches. RNA sequencing was performed and associated mechanistic signaling pathways were analyzed based on the Kyoto Encyclopedia of Genes and Genomes data. The analytical results of RNA sequencing data were further validated by in vitro and in vivo experiments. **Results:** Withaferin A reduced the I/R injury-induced apoptotic death of HRMECs in vitro with a good drug-like property. RNA sequencing and experimental validation results indicated that withaferin A increased the production of the crucial antioxidant molecules heme oxygenase 1 (HO-1) and peroxiredoxin 1 (Prdx-1) during I/R. In addition, withaferin A activated the Akt signaling pathway and increased the expression of HO-1 and Prdx-1, thereby exerting an antioxidant effect, attenuated the retinal I/R injury, and decreased the apoptosis of HRMECs. The blockade of Akt completely abolished the effects of withaferin A. **Conclusions:** The study identified for the first time that withaferin A can protect against the I/R-induced apoptosis of human microvascular retinal endothelial cells via increasing the production of the antioxidants Prdx-1 and HO-1. Results suggest that withaferin A is a promising drug candidate for the treatment of retinal I/R injury.

## 1. Introduction

Retinal ischemia-reperfusion (I/R) injury is a pathological process which contributes to a number of retinal illnesses, including glaucoma, diabetic retinopathy, and retinal vascular occlusion. These eye diseases are considerable causes of vision loss and irreversible blindness worldwide [1,2]. Characterized by primary ischemia and the overproduction of reactive oxygen species (ROS) from reperfusion-reoxygenation injury, oxidative stress contributes to the progression of retinal I/R, and triggers the irreversible damage or death of human retinal capillary endothelial cells [3,4]. The tight connections of neighboring retinal capillary endothelial cells make up the inner blood–retinal barrier (iBRB) [5,6]. During in the early stage of retinal I/R, the death of human retinal microvascular endothelial cells (HRMECs) destroys the iBRB integrity and thereby increases retinal capillary permeability, and thus is associated with the cascades of neurotic retinal injury, retinal edema, and vision loss [7,8]. Therefore, the amelioration of oxidative stress in retinal capillary endothelial cells may serve as an important treatment approach in the early stage of retinal I/R.

Active compounds extracted from herbs have been safely used to treat various ailments worldwide owing to the belief of there being fewer side effects of naturally occurring compounds [9]. The first chemical of the withanolide group identified from the medicinal herb Withania somnifera was withaferin A [10,11]. As withaferin A possesses antibacterial and antiinflammatory properties, it has been utilized for generations in traditional medicine. [12]. The efficacies of withaferin A in the treatment of prostate, breast, and pancreatic cancers have been recently reported [10]. Notably, we found that withaferin A can inhibit apoptosis by inhibiting oxidative stress in cardiac I/R [13]. A recent study indicated that withaferin A can protect endothelial cells from oxidative stress [14]. However, to our knowledge, it is still unclear if withaferin A exerts an antioxidant effect on retinal capillary endothelial cells, thereby protecting against retinal I/R injury. The present study investigated the potential protective effects of withaferin A on retinal I/R injury via inhibiting oxidative stress.

## 2. Methods

### 2.1. Drug-Likeness Prediction of Withaferin A

The drug-likeness of withaferin A was evaluated using the SwissADME web tool (http://www.swissadme.ch/ (accessed on 16 October 2020). This web tool can be employed to ascertain the unique ADME behaviors of the natural compounds from plants and is accessible on a web server that shows the submission page of SwissADME in Google. The molecular sketcher of a compound can be shown in the input zone based on Chem Axons Marvin JS (http://www.chemaxon.com (accessed on 16 October 2020), thus enabling users to draw and edit the 2D chemical structure of the compound. The bioavailability radar allows users to preview the drug-likeness of the compounds of interest. The six axes of the Bioavailability Radar represent six important properties for bioavailability. LIPO (lipophilicity), SIZE, POLAR (polarity), INSOLU (insolubility), INSATU (insaturation), and FLEX (flexibility) are the six physicochemical characteristics that are taken into account. Each property is defined by a descriptor of SwissADME, and a range of optimal values is highlighted in pink. The red line of the compound under study must be fully included in the pink area; any deviation represents a suboptimal physicochemical property for oral bioavailability. Drug-likeness prediction focuses on the rapid screening of chemical libraries to select the best molecules that can be developed into drugs [15].

### 2.2. Cell Culture and Treatment

Primary human retinal microvascular endothelial cells (HRMECs, Cat# cAP-0010, Angio Proteomie, Boston, MA, USA), passages 2–5, were cultured in endothelial basal media containing 5% fetal bovine serum and premixed endothelial cell growth supplement. Upon reaching 100% confluence, cells were serum starved for 3 h and then treated with 1% fetal bovine serum.

To simulate retinal I/R injury at the cellular level, HRMECs were cultured in plates and subjected to hypoxia (1% O_2_) for 3, 6, 12, or 24 h in D-Hanks buffer followed by 2 h of reoxygenation (5% O_2_) in complete DMEM. To further study the impact of I/R-induced oxidative stress, cultured HRMECs were subjected to 2 h of hydrogen peroxide (H_2_O_2_) at different concentrations (50, 100, 200, 400, or 800 µmol/l). At the same time, the effects of Withaferin A were tested. Withaferin A was purchased from ChromaDex (Cat# 5119-48-2, Irvine, CA, USA).

### 2.3. Cell Survival Assay

Cell vitality assay was performed using the MTT Cell Proliferation Assay kit (Cat#CT01-5, Sigma-Aldrich, St. Louis, MO, USA) following the manufacturer’s instructions. Cell cytotoxicity assay was performed using a lactate dehydrogenase (LDH) cytotoxicity assay kit (Cat#MK401, Takara Shuzo Co., Shiga, Japan).

### 2.4. Cell Apoptosis Assay

Apoptotic cell death was determined using a V-FITC Apoptosis Detection kit (Cat#a211-01, vazyme, Nanjing, China) following the manufacturer’s instructions. Cell staining was conducted with fluorescein isothiocyanate (FITC)-labeled Annexin V (green) and propidium iodide (PI; red), according to our previous study [13].

### 2.5. Western Blotting

HRMECs and mice retinal tissues were lysed using lysis buffer (Cat# 9803, Cell Signaling Technology, Topsfield, MA, USA). The total protein (50 μg) was loaded to the Bio-Rad gel (4–20%) and subjected to electrophoresis, then was transferred to a polyvinylidene fluoride membrane, and immunoblotted with primary antibodies against serine 473 of phosphorylated Akt (Cat# 9271), total Akt (Cat#4691), phosphorylated ERK (Thr202/Tyr204) (Cat# 9106), total ERK (Cat# 9107), heme oxygenase 1 (HO-1; Cat# 70081), peroxiredoxin 1 (Prdx-1; Cat# 8499), cleaved caspases-3 (Cat# 9661), total caspase-3 (Cat# 9662), β-actin (Cat#SC-4970), and the secondary antibodies (Cat#58802 and Cat#91196, all antibodies were purchased from Cell Signaling Technology, Danvers, MA, USA). After being treated with Super-Signal reagent (Cat#34095, Pierce, IL, USA), the membrane was photographed using a Bio-Rad ChemiDoc Touch station (Bio-Rad Laboratories, Hercules, CA, USA).

### 2.6. RNA Sequencing and Analysis

Differential gene expression analysis was performed using RNAseq at Shanghai Biotree Biological Technology [16]. HRMECs were incubated with withaferin A (50 nmol/mL) for 15 min followed by I (3 h)/R (2 h) treatment. Total RNA was isolated using the mirVanamiRNA Isolation Kit (Cat# AM1561, Ambion, Austin, TX, USA) following the manufacturer’s instruction. The Agilent 2100 Bioanalyzer was used to assess the integrity of the RNA (Agilent Technologies, Santa Clara, CA, USA). The samples that had an RNA Integrity Number (RIN) of less than seven were used in the analysis. The mRNA library was created using the TruSeq StrandedmRNA LTSample Prep Kit (Cat# RS-122-2101, Illumina, San Diego, CA, USA). Then, 150 bp paired-end reads were produced using the Illumina sequencing platform (HiSeqTM 2500 or Illumina HiSeq X Ten). Using the DESeq2R software (1.16.1), differential expression analysis was carried out. Using a model based on the negative binomial distribution, DESeq2 was employed to develop statistical algorithms for identifying differential expression among digital gene expression data. The resulted *p*-values were modified using Benjamini and Hochberg’s method for reducing the false discovery rate. Genes identified by DESeq2 as having differential expression were those with an adjusted *p*-value 0.05. KEGG is a database of biological systems that combines information on the genetic, chemical, and systemic functioning aspects of living things (http://www.genome.jp/kegg/ (accessed on 22 August 2021). Based on molecular-level data, particularly that from sizable molecular datasets produced by genome sequencing and other high-throughput experimental technologies, KEGG offers a reference knowledge base for comprehending the high-level functions and utilities of biological systems, such as cells and organism ecosystems. Using the clusterProfiler R package, the statistical enrichment of differently expressed genes in KEGG pathways was carried out.

### 2.7. Quantitative PCR

Primers were purchased from Integrated DNA Technologies. Cells were used to extract total RNA using a RNeasy Mini kit (Cat# 74106, QIAGEN, Hilden,, Germany). Quantitative PCR was performed using a 7900HT Fast Real-Time PCR System (Applied Biosystems, Foster City, CA, USA). The thermal cycling conditions were as follows: denaturation at 95 °C for 5 min, followed by 40 cycles of 10 s at 95 °C, 20 s at 55 °C, and 20 s at 72 °C.

### 2.8. Measurement of ROS Generation

Inverse fluorescence microscopy (Olympus, Japan) was used to take pictures of the stained cells after they had been incubated with 2 M dihydroethidium (DHE) (Cat# D23806, Thermo-Fisher Scientific, Pittsburgh, PA, USA) for 10 min. To measure retinal ROS levels, 3-nitrotyrosine protein levels were determined in mice retinal lysates using an Oxiselect Nitrotyrosine kit (Cat#STA-305, Cell Biolabs, San Diego, CA, USA) according to the manufacturer’s instructions.

### 2.9. Animal Experiments and Establishment of Retinal I/R Model

All procedures in this study were approved by the Shanxi medical university IACUC Committee on Animal Care (2021-047) and adhered to both the guidelines set forth by the National Institute of Health’s Guide for the Care and Use of Laboratory Animals and the Association of Research in Vision and Ophthalmology (ARVO) statement for the use of animals in ophthalmic and vision research. Male C57Bl/6J mice aged eight weeks (18–22 g) were bought from Cyagen Biosciences Inc. (Guangzhou, China). Mice were kept in temperature-controlled environments with a 12 h light/dark cycle for two weeks. Six mice in each group were used for the in vivo experiment. Tissue samples were collected after animal euthanasia by 2% isoflurane anesthesia.

To establish the retinal I/R model, mice were unilaterally ligated on both the left pterygopalatine artery and left external carotid artery for 3 h to induce retinal ischemia and then the arteries were released for 24 h to create reperfusion. Sham surgery was conducted on the right side of mice served as autologous control [17]. Double-distilled water or 10 nmoles/g withaferin A was intraperitoneally injected 8 h before surgery. To evaluate the role of Akt in the protective effect of withaferin A, an Akt inhibitor (MK2206; 4 μg/g) was administrated via intraperitoneal injection 30 min prior to withaferin A injection.

### 2.10. Determination of Retinal Caspase-3 Activity

The harvested retinas were immediately sonicated on ice. The caspase-3 activity in the retinal supernatants was assessed. Promega Assay (Madison, WI, USA), which supplied a Fluorometric CaspACE assay system, was carried out in duplicate for each sample, together with the required blanks and negative controls, following the manufacturer’s instructions. With each group of unknowns, 7-amino-4-methyl coumarin (AMC) was measured, and standard curves were created. Caspase-3 activity was assessed in terms of fluorescence and picomoles of AMC released per milligram of protein per minute.

### 2.11. Statistical Analysis

All values are expressed as mean ± standard error of mean (SEM) of independent experiments. One-way ANOVA was performed, followed by a Bonferroni post hoc analysis. Image Lab 5.2 was used to measure the Western blot densities (Bio-Rad, Hercules, CA). GraphPad Prism Software 8.0 was used to conduct the statistical analysis. Statistical significance was defined as *p* < 0.05.

## 3. Results

### 3.1. Withaferin A Possesses Good Drug-like Properties and Reduces Retinal I/R Injury and H_2_O_2_-Induced Death of HRMECs

The drug-likeness of withaferin A was evaluated using the SwissADME web tool [18]. The physicochemical properties and structural features suggested that withaferin A is a promising drug candidate with value of developing to a drug (Figure 1).

Next, HRMECs were subjected to ischemia for different time periods, followed by 2 h of reperfusion (Figure 2A). The protective effects of withaferin A at graded concentrations were observed at the same time. Cell survival assays (MTT and LDH) showed that withaferin A ameliorated the decline of cell survival induced by I/R in a dose-dependent manner (Figure 2B and E).

To further confirm the protective effects of withaferin A, H_2_O_2_ was used to simulate oxidative stress damage in retinal I/R (Figure 2C), and MTT assay was used to determine cell viability. As expected, treatment with withaferin A effectively and dose-dependently recovered the cell viability that was decreased by H_2_O_2_ exposure (100 μmol/L, 2 h) (Figure 2D). In addition, LDH assay revealed that H_2_O_2_-induced cell death was reduced by withaferin A in a dose-dependent manner (Figure 2F). These results demonstrated that withaferin A effectively rescued retinal I/R injury and H_2_O_2_-induced HRMEC death.

### 3.2. Withaferin A Inhibits HRMECs Apoptosis

To further determine whether cell necrosis and/or cell apoptosis were involved in the protective effect of withaferin A, we used z-VAD-FMK to inhibit caspase activity. Results showed that z-VAD-FMK effectively blocked withaferin A-induced reduction of LDH release (Figure 3A), suggesting that the protective effect of withaferin A was mainly due to inhibition of apoptosis. To further confirm this finding, apoptotic cell death was analyzed using a V-FITC apoptosis detection kit and Western blotting. As illustrated in Figure 3B and 3C, withaferin A significantly decreased the early (Annexin V^+^/PI^−^) and late stages of apoptosis (Annexin V^+^/PI^+^) induced by H_2_O_2_. Moreover, withaferin A strongly reduced the appearance of H_2_O_2_-induced apoptotic caspase−3 cleaved fragments in HRMECs (Figure 3D). Taken together, these findings indicated that withaferin A could reduce I/R injury-induced apoptotic cell death of HRMECs.

### 3.3. Withaferin A Protects against I/R Injury of HRMECs by Increasing the Levels of Antioxidant Heme Oxygenase 1 (HO-1) and Peroxiredoxin 1 (Prdx-1)

Nonbiased RNA sequencing was performed to explore the molecular mechanisms by which withaferin A reduces I/R-induced apoptosis. Retinal endothelial I/R injury was simulated in HRMECs and the effects of withaferin A (50 nmol/mL) were observed. Differently expressed genes (DEGs) were determined using RNAseq. Up to 1164 genes showed significant upregulation after treatment with withaferin A (Figure 4A,B). Significantly enriched signaling pathways were analyzed using the KEGG approach (Figure 4C). Results showed that withaferin A substantially upregulated the expression of 81 genes (fold change > 2). As some of these genes showed low expression in HRMECs, we focused on 12 of the 81 genes (FPKM > 3). Results showed that withaferin A reduced HRMEC apoptosis by stimulating the expression of the 12 genes (Figure 4D). Among the 12 genes (Appendix A), 9 genes in HRMECs were detected using quantitative PCR (Figure 4E), of which HMOX1 was the most significantly increased gene in expression. Moreover, withaferin A remarkably increased the expression of PRDX1 gene. It is known that HMOX1 and PRDX1 play a pivotal role in inhibiting oxidative stress [19,20]. Our results suggest that withaferin A exerts an antioxidant effect in the retina by upregulating the expression of HO-1 and Prdx-1.

### 3.4. Withaferin A Increases the Expressions of HO-1 and Prdx-1 via Activating Akt

Western blotting results (Figure 5A) showed that the expression of Prdx-1 and HO-1 were significantly increased by withaferin A (50 nmol/mL for 8 h). To explore the signaling by which withaferin A upregulates HO-1 and Prdx-1 expressions, we searched for the KEGG analysis results, and found that signal transduction was the most significantly enriched signaling pathway of withaferin A (Figure 4C). Accordingly, we screened the upstream signaling that controls the expression of HO-1 and Prdx-1. Although several studies have reported that Akt and ERK are the upstream signals of Prdx-1 and HO-1 [21,22], whether the Prdx-1 and HO-1 upregulation observed in our study was associated with Akt and/or ERK was unclear. We found that pretreatment of HRMECs with withaferin A for 15 min and 30 min significantly increased the phosphorylation of Akt and ERK (Figure 5B). To further demonstrate the cause-and-effect relationship among Akt, ERK, HO-1 and Prdx-1, we pretreated HRMECs with ERK inhibitor U0126 (10 μM) for 15 min and with Akt inhibitor MK2206 (1 μM) for 15 min, then observed the expression of Prdx-1 and HO-1. Results showed that MK2206 significantly inhibited the withaferin A-induced upregulation of Prdx-1 and HO-1 (Figure 5C). However, U0126 did not affect the expression of Prdx-1 and HO-1 (Figure 5C). Taken together, these results provided clear evidence that withaferin A increases Prdx-1 and HO-1 expression via the Akt-dependent signaling pathway.

### 3.5. Withaferin A Inhibits H_2_O_2_-Induced Apoptosis of HRMECs via the Akt/antioxidant Signaling Pathway

ROS-induced oxidative stress is involved in the progression of retinal I/R injury [23]. To further confirm that treatment with withaferin A attenuated cell apoptosis during retinal I/R injury via the Akt/antioxidant signaling pathway, HRMECs were pretreated with the inhibitors for 15 min before the addition of 50 nM withaferin A. HRMECs were then incubated with H_2_O_2_ (50 μmol/L) for 8 h and ROS generation was measured. We found that withaferin A reduced H_2_O_2_-induced ROS generation. MK2206, but not U0126, blocked the ROS-suppression effect of withaferin A (Figure 6A,B). Moreover, as indicated in Figure 6C, withaferin A ameliorated the cell viability lowered by H_2_O_2_, whereas pretreatment with MK2206 inhibited this effect. Western blotting was used to determine the role of the Akt/antioxidant signaling pathway in the anti-apoptotic effect of withaferin A. As summarized in Figure 6D, withaferin A increased Prdx-1 and HO-1 expression; however, these effects were blocked by pretreatment with MK2206. Cleaved caspase-3 levels were significantly decreased after treatment with withaferin A in H_2_O_2_-induced cell injury; however, this effect was blocked by treatment with MK2206. Taken together, these results suggested that withaferin A could prevent H_2_O_2_-induced cell apoptosis via the Akt-dependent Prdx-1/HO-1/antioxidant signaling pathway.

### 3.6. Withaferin A Attenuated Retinal I/R Injury In Vivo via the Akt-Dependent Prdx-1 and HO-1 Signaling Pathway

To confirm whether withaferin A could protect against retinal I/R injury in vivo via the Akt signaling pathway mentioned above, 4 μg/g of MK2206 was injected intraperitoneally 30 min prior to the administration of withaferin A (10 nmoles/g), and then retinal I/R surgery was performed 8 h later. Retinal Prdx-1 and HO-1 expressions were measured in each group. As expected, administration of withaferin A increased Prdx-1 and HO-1 expression, whereas this effect was blocked after treatment with MK2206 in vivo (Figure 7A). Moreover, even though withaferin A blocked I/R-mediated nitrotyrosine expression, it decreased caspase-3 activity marginally, whereas the effects were inhibited by MK2206 (Figure 7B,C). These in vivo findings preliminarily verified that withaferin A attenuates retinal I/R injury via activating the Akt-dependent Prdx-1/HO-1 antioxidant signaling pathway.

## 4. Discussion

This study investigated the protective effect of withaferin A on retinal I/R injury and analyzed the underlying signaling mechanisms. We found that withaferin A could stimulate the Akt signaling pathway in the retina, enhance HO-1 and Prdx-1 expression, thus exert an antioxidant effect and attenuated retinal I/R injury by decreasing the apoptosis of retinal capillary endothelial cells. Blockade of Akt, but not ERK, completely abolished the effects of withaferin A described above. The findings from this study suggest that withaferin A, a naturally occurring phytochemical, could be a drug candidate to alleviate retinal I/R injury.

This study presented several novel observations. First, we demonstrated that, as a small molecule compound of herbal origin, withaferin A has the potential to be developed as a drug at a later stage. Before the development of synthetic drugs, natural products were used as an alternative therapy to treat diseases [24,25]. Withaferin A is a bioactive withanolide extracted from the medicinal plant *Withania somnifera* [10,11] and has been safely used for centuries [12]. It has attracted considerable research attention, as it is associated with unique properties and pharmacological activities [9]. The SwissADME web tool used for analysis [26] revealed the good drug-like properties of withaferin A and highlighted this compound as a potential candidate for further development as a drug. These findings make the research on withaferin A more meaningful.

Second, we demonstrated for the first time that withaferin A could reduce the apoptotic death of retinal capillary endothelial cells after retinal I/R injury. Retinal capillary endothelial cells are the basic components of the iBRB that play an important role in maintaining the internal environmental homeostasis of the retina and in its recovery after I/R [4,5]. Capillary endothelial cell apoptosis induced by retinal I/R triggers cascades leading to leakage of the capillary vascular fluid and causes retinal edema, tissue damage, neurotic retinal injury, and vision loss [27]. Thus, it is beneficial to suppress endothelial cell apoptosis at the early stages of retinal I/R injury [4,28]. Our previous study showed that withaferin A effectively downregulated the I/R-induced apoptosis of cardiomyocytes [13]. However, it is unknown whether withaferin A exerts similar effects in retinal endothelial cells. Our findings from this study provide direct evidence that withaferin A indeed effectively rescues retinal capillary endothelial cell death after I/R injury, and this effect was blocked by the caspase inhibitor z-VAD-FMK, suggesting that the protective effect of withaferin A on retinal I/R injury was targeted toward cell apoptosis.

Third, withaferin A has an antioxidant effect in retinal I/R. Based on RNA-seq results, we focused on the top 9 genes among the 81 genes that were remarkably upregulated by withaferin A during I/R. PCR indicated a significant increase in HMOX1 and PRDX1. HMOX1 and PRDX1HO-1 and Prdx-1 are antioxidant genes [29] known to regulate the protein expression of HO-1 and Prdx-1. Prdx-1 is one of the guardians against oxidative stress and a modulator of peroxide signaling [30], whereas HO-1 protects several tissues and organs from oxidative stress [31]. Our findings suggest that withaferin A could upregulate the expression of HO-1 and Prdx-1, and consequently, mediate the antioxidant pathway in retina.

Fourth, we elucidated a novel cellular mechanism by which withaferin A directly ameliorates retinal I/R injury. As the KEGG signaling pathway analysis of RNAseq results identified signal transduction as the most significantly enriched signaling pathway of withaferin A, we explored the upstream signaling of HO-1 and Prdx-1 regulation. By inhibiting the activities of ERK and Akt (the possible upstream regulators of HO-1 and Prdx-1) using U0126 and MK2206, we provided direct evidence that the antiapoptotic and antioxidant effects of withaferin A on HRMECs were mediated by the Akt pathway, because inactivation of Akt abolished the observed protective effects of withaferin A. Furthermore, due to the complex three-dimensional structure of the retina consisting of 10 layers and multilayered blood vessels, we used whole retinal homogenates for preliminary verification the protective effects and molecular mechanisms of withaferin A in vivo [32,33]. We found that withaferin A increased the expression of Prdx-1 and HO-1, meanwhile, it blocked I/R-mediated nitrotyrosine (ROS derivative) expression and decreased caspase-3 activity marginally in retinal homogenates. However, the effects of withaferin A mentioned above were inhibited by MK2206 in vivo. Together, we preliminarily verified that withaferin A ameliorated retinal I/R injury by reducing apoptosis via the Akt-dependent Prdx-1 and HO-1 antioxidant signaling pathways in a mouse model of retinal I/R.

## 5. Conclusions

The present study demonstrated that withaferin A, a small and easily extractable phytochemical, can ameliorate retinal I/R injury by reducing endothelial cell apoptosis via the Akt-dependent Prdx-1 and HO-1 antioxidant signaling pathways. Withaferin A has the potential to be a drug candidate in the treatment of retinal I/R injury.

## 6. Limitations

As cell survival and metabolism are linked to the Akt signaling pathway [34,35], we conducted in vivo and in vitro experiments using pharmacological methods but not gene engineering model to further understand the molecular mechanisms of withaferin A. Second, the protective effect of withaferin A on retinal I/R injury may involve other signaling pathways. For example, a study reported withaferin A as a potential antiangiogenic agent in retinal neovascularization, which is a well-known late pathological change in retinal I/R injury [36]. However, it was not possible to concentrate on multiple paths at once. Thus, further clarification on the effect of withaferin A in the late stages of retinal I/R injury is warranted.

## Figures and Tables

**Figure 1 cells-11-03113-f001:**
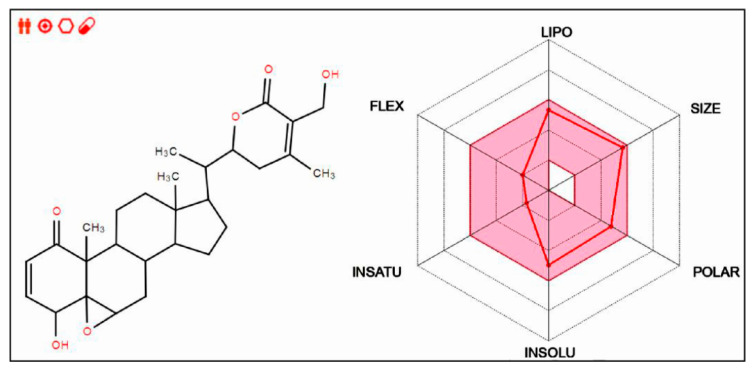
The chemical structure and drug-likeness of withaferin A identified by SwissADME web tool. Left, chemical structure of withaferin A. Right, radar graph showing the bioavailability of withaferin A (detailed information can be seen in Methods section of the main text).

**Figure 2 cells-11-03113-f002:**
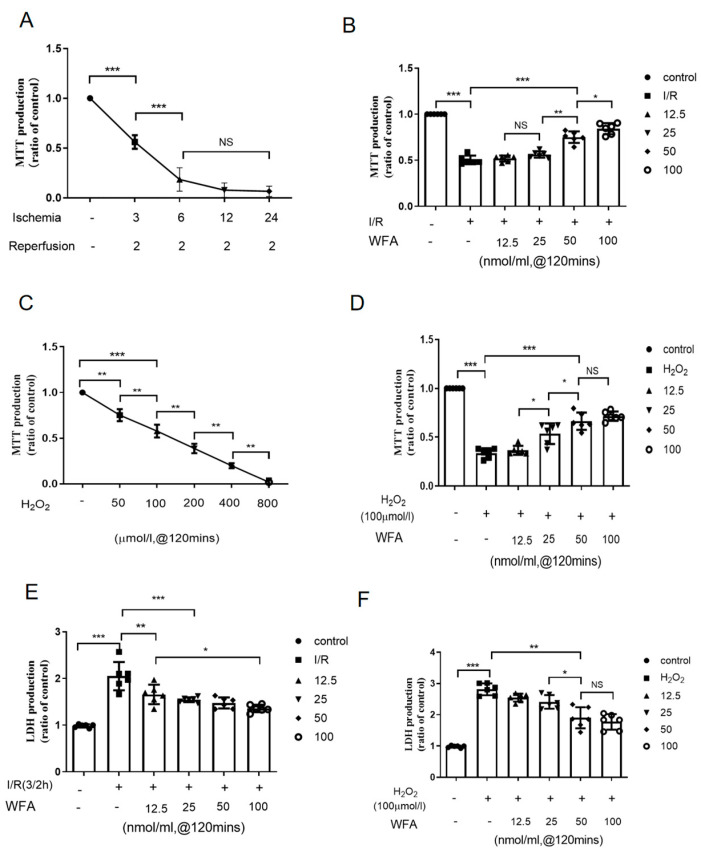
Withaferin A reduced retinal I/R injury and H_2_O_2_-induced death of HRMECs. (**A**–**D**) MTT assays of HRMECs at different treatments. (**E**,**F**) LDH assays of HRMECs at different treatments. Data are expressed as mean ± SEM. * *p* < 0.05, ** *p* < 0.01, *** *p* < 0.001. N = 6 for each treatment.

**Figure 3 cells-11-03113-f003:**
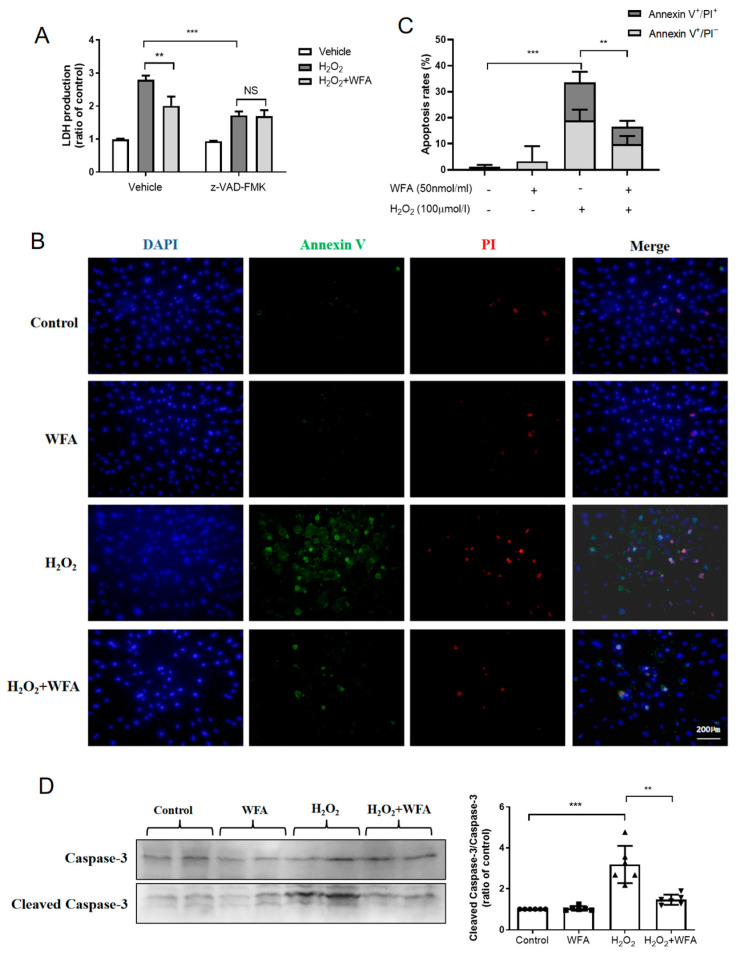
Withaferin A reduced H_2_O_2_-induced apoptotic death of HRMECs. HRMECs were pretreated with withaferin A (50 nmol/mL for 15 min) followed by treatment with H_2_O_2_ (100 μmol/L) for 2 h. Pretreatment of HRMECs with z−VAD-−FMK (20 μM, 1 h) significantly inhibited the protective effect of withaferin A on cell apoptotic death. (**A**) LDH assay and apoptotic rate. (**B**,**C**) Immunostaining of HRMECs for Annexin V (green), PI (red), and DAPI (blue). It is a method for the identification of apoptotic cells (Annexin V^+^/PI^−^, the early of apoptosis; Annexin V^+^/PI^+^, the late stages of apoptosis). (**D**) Western blots and quantification of cleaved caspase−3 in HRMECs. Two strips as a group, withaferin A inhibited the H_2_O_2_-induced apoptotic caspase−3 cleaved fragments. Results are expressed as the mean ± SEM. ** *p* < 0.01, *** *p* < 0.001; NS, no significance. N = 6 repeated experiments.

**Figure 4 cells-11-03113-f004:**
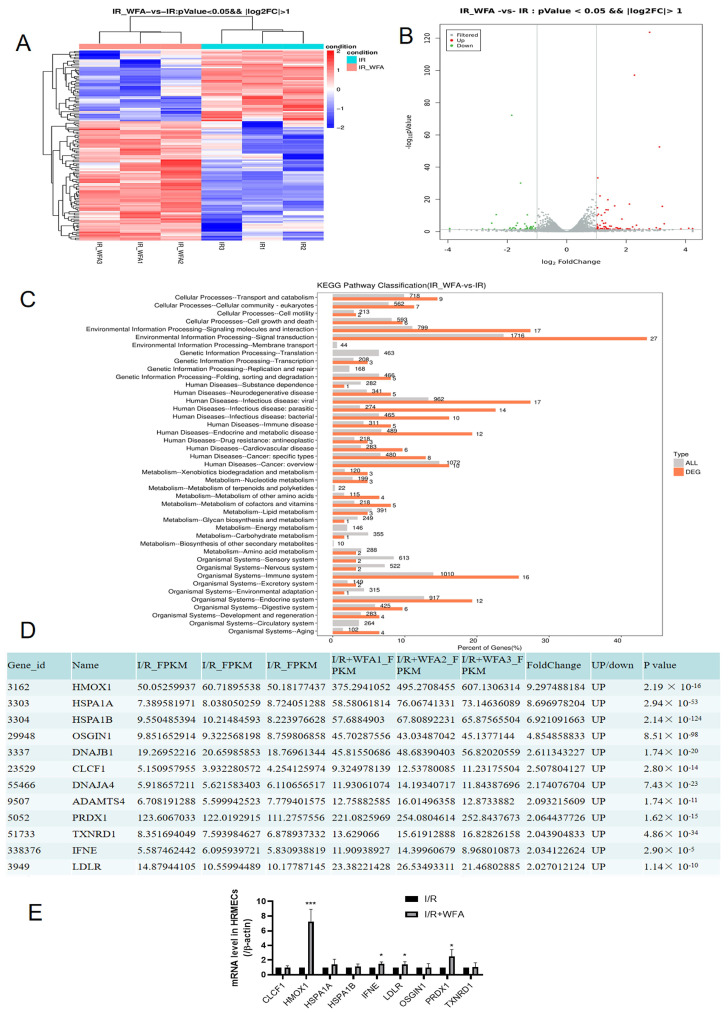
RNA−sequencing results to demonstrate the molecular mechanisms by which withaferin A protected against HRMEC I/R injury. HRMECs were treated with withaferin A (50 nmol/mL) or untreated (control) for 15 min, followed by treatments of simulating ischemia (3 h)/reperfusion (2 h). (**A**) Cluster analysis of DEGs. (**B**) Volcano plot of RNA−seq. Downregulated genes induced by withaferin A were represented by green dots with a value of log2FoldChange < −1. Upregulated genes induced by withaferin A shown by orange dots with log2FoldChange values > 1. (**C**) KEGG pathways that were significantly affected by withaferin A in HRMECs. (**D**) The relative abundance of 12 genes were considered acceptable among the 81 significantly upregulated genes induced by withaferin A (fold change > 2, FPKM > 3). N = 3, *p* values less than 0.05 were considered significant. (**E**) Real−time PCR validation on the RNA−seq results (total 9 genes examined). N = 3 repeated experiments/conditions. Results are expressed as mean ± SEM. * *p* < 0.05, *** *p* < 0.001.

**Figure 5 cells-11-03113-f005:**
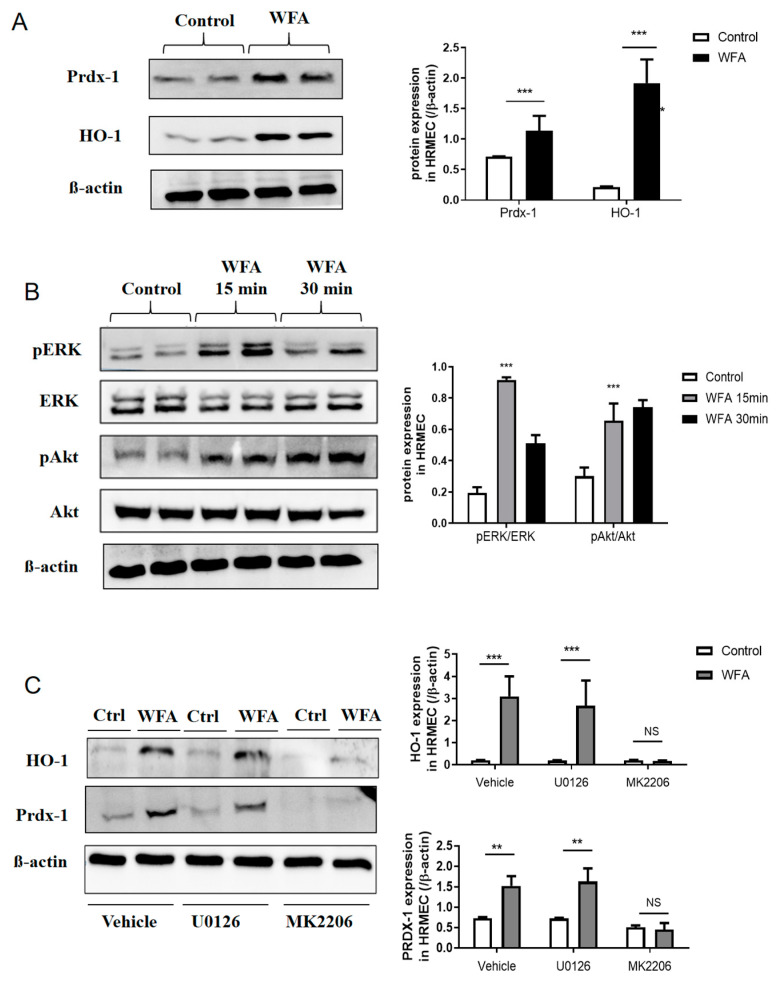
Role of Akt in withaferin A-induced HO-1 and Prdx-1 expression in HRMECs. (**A**) Incubation of HRMECs with withaferin A (50 nmol/mL, 8 h) enhanced HO-1 and Prdx-1 expression. (**B**) Representative immunoblots showing ERK and Akt activation (phosphorylation) in HREMCs treated with withaferin A. (**C**) MK2206 (1 μM) or U0126 (10 μM) treatment for 15 min before treatment with withaferin A. Western blots showing that MK2206 significantly blocked the withaferin A-mediated upregulation of HO-1 and Prdx-1 expression. N = 6 repeated experiments/conditions. All results are expressed as the mean ± SEM. ** *p* < 0.01, *** *p* < 0.001. NS, no significance.

**Figure 6 cells-11-03113-f006:**
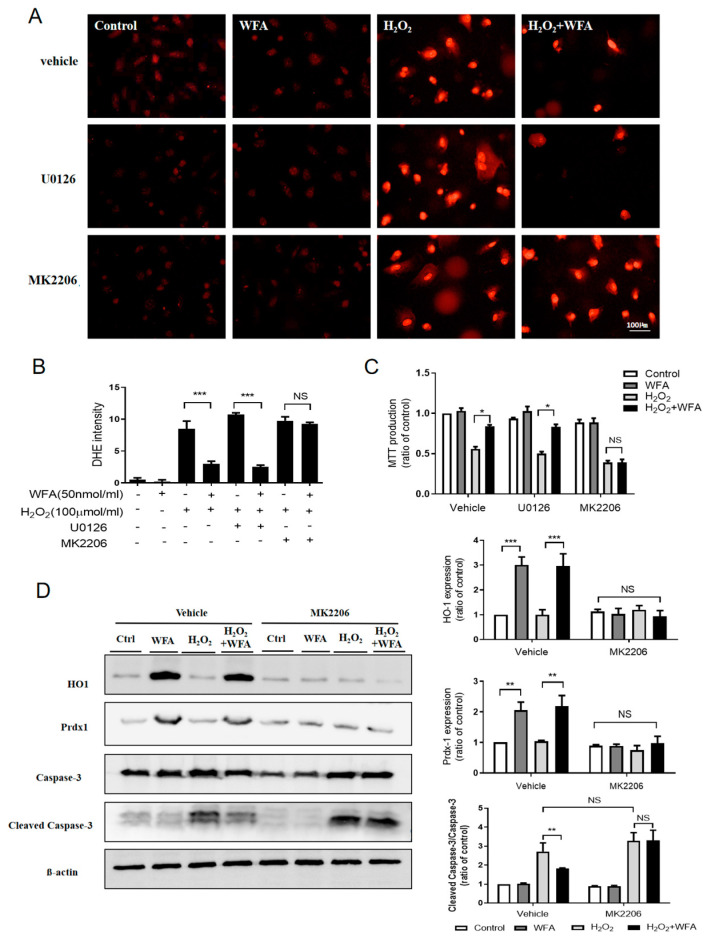
Withaferin A suppressed H_2_O_2_-induced cell apoptosis via the Akt−dependent Prdx−1 and HO−1 antioxidant signaling pathway. HREMCs were pretreated with U0126 or MK2206 for 15 min before treatment with withaferin A (50 nmol/mL) for 15 min, followed by incubation with H_2_O_2_ (50 μmol/L) for 8 h. (**A**,**B**) ROS generation was determined using DHE staining. (**C**) MTT cell survival assay to evaluate cell viability. (**D**) Western blotting to quantify cleaved caspase−3, caspase−3, HO−1, and Prdx−1 protein expression. Results were expressed as mean ± SEM. * *p* < 0.05, ** *p* < 0.01, *** *p* < 0.001. N = 6 for each treatment.

**Figure 7 cells-11-03113-f007:**
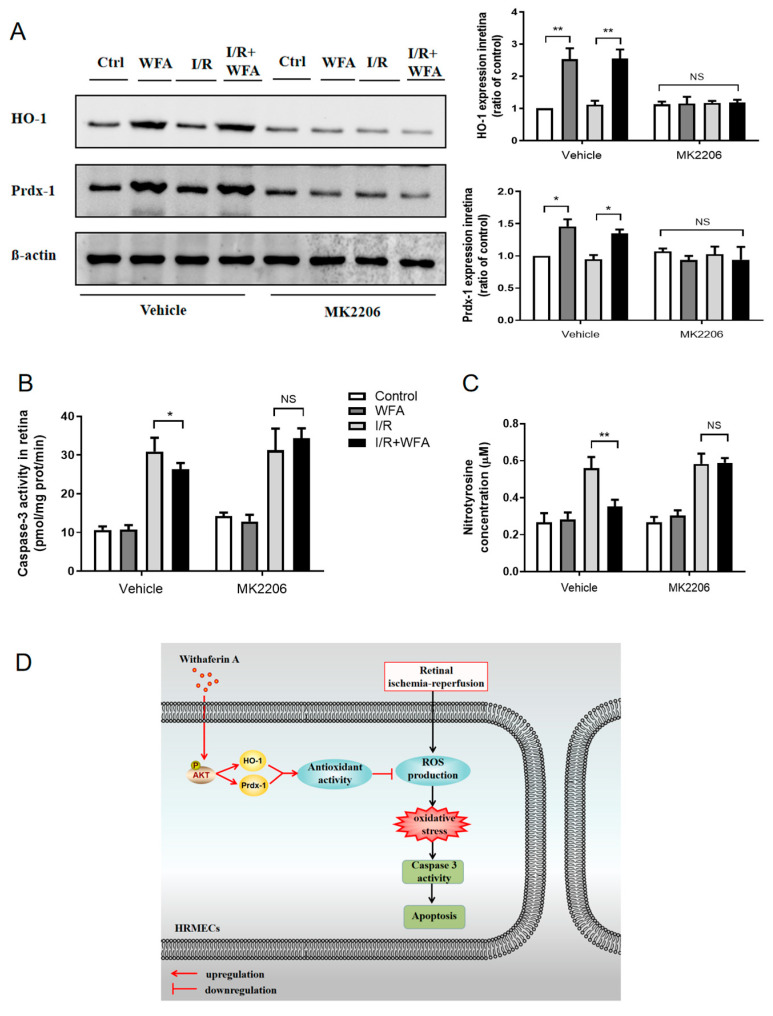
Inhibition of Akt blocked the Prdx-1 and HO-1 antioxidant signaling pathways in retinal I/R injury in vivo. (**A**) Western blotting to quantify the retinal protein expressions of HO-1 and Prdx-1 in mice treated with the vehicle or with I/R injury, with or without treatment with withaferin A (10 nmoles/g) and MK2206 (4 μg/g). (**B**) Quantification of caspase-3 activity in the retina. (**C**) Quantification of nitrotyrosine (ROS derivative) in the retina. (**D**) Diagram depicting the mechanism responsible for the protective effects of withaferin A on retinal I/R injury. Results are expressed as mean ± SEM. * *p* < 0.05, ** *p* < 0.01; NS, no significance. N = 6 for each treatment.

## Data Availability

Not applicable.

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
