# Peer review of "Withaferin a Attenuates Retinal Ischemia-Reperfusion Injury via Akt-Dependent Inhibition of Oxidative Stress"

_cells, 2022, doi:10.3390/cells11193113_

Round 1
Reviewer 1 Report
Dear Authors,
It's a great focus on the subject but i would like to tell you that this review needs modifications before considered, please see the following list of points which need to be considered:
- Annexin V and ROS must be quantified by FACS, as the reduction is visualized in microscopy, but cannot be quantified.
-The abstract is not well organized to reach to conclusions
-The introduction need more organization
My Best Regards
-Inapparent key points in the paper make it unique
Author Response
Point-to-point Responses to Reviewer #1
Comments and Suggestions for Authors
It's a great focus on the subject but i would like to tell you that this review needs modifications before considered, please see the following list of points which need to be considered:
Response: We greatly appreciate this reviewer’s positive assessment on our manuscript and agree with all his/her expert suggestions. Specifically, we have made the following revisions.
Q1. Annexin V and ROS must be quantified by FACS, as the reduction is visualized in microscopy, but cannot be quantified.
A1. We apologize our original figures were too terse and lacked details described quantification of Annexin V and ROS, which may cause the misunderstanding. We performed semi-qualifications of apoptotic rate by counting the the early (Annexin V+/PI−) and late stages of (Annexin V+/PI+) cells apoptosis and the DHE staining for ROS level in multi-fields (1,2) (n=5 fields for each slide and total 4 slides for each treatment). The results have been shown in Figure 3C and Figure 6B in the revised version. To further confirm the results, we had performed Western blotting experiments on caspase3 and antioxidant Prdx-1 and HO-1. We agree with the reviewer that the better way of quantification is FACS. However, we are unable to finish the FACS experiments before the deadline of revision submission (total 10 days).Therefore, we hope semi-qualifications could be acceptable.
Q2. The abstract is not well organized to reach to conclusions.
A2. As requested, we revised the abstract.
Q3. The introduction need more organization
A3. We revised the Introduction to make it more organized.
Q4. Inapparent key points in the paper make it unique
A4. As requested, we added “phytochemical” to the key points.
- Guan H, Zhang H, Cai J, et al. IKBKE is over‐expressed in glioma and contributes to resistance of glioma cells to apoptosis via activating NF‐κB[J]. The Journal of pathology, 2011, 223(3): 436-445.
- Liu H, Mo H, Yang C, et al. A novel function of ATF3 in suppression of ferroptosis in mouse heart suffered ischemia/reperfusion[J]. Free Radical Biology and Medicine, 2022, 189: 122-135.
Reviewer 2 Report
The study investigated the effect of withaferin A, an active compound isolated from plants, in oxidative stress-mediated death of HRMECs. The authors investigated its antioxidant effect using hypoxia and hydrogen peroxide in vitro as well as a mouse model of ischemia/reperfusion injury. Withaferin A protects against cell apoptosis by increasing HO-1 and Prdx-1 expression via AKT phosphorylation. The in vitro mechanism study is solid, and the finding is interesting. However, in vivo data could not provide any evidence for the therapeutic possibilities against retinal endothelial cell protection since they used whole retinal homogenates. The author did not assess how much withaferin A reaches to the retina after systemic injection. (e.g. What is the rationale of 10 mmol/g bw?). Whole retinal flat mount immunostaining can be a useful tool to evaluate the apoptosis of retinal endothelium. Furthermore, the manuscript is not fully ready to publish. The authors are encouraged to edit the manuscript carefully. E.g. revise enormous errors in a text such as abbreviations, the concentration of chemicals, statistical analysis, and lack of information in the legends. There are representative concerns:
1. Methods: missing catalog numbers, city, and country in some reagents, assay kits, secondary antibodies, and primer sequences.
2. Method 2.9: note animal numbers, anesthesia, ARVO Statement for the Use of Animals in Ophthalmic and Vision Research, IACUC protocol number
3. Fig. 1: did not interpret what does radar graph means, any positive control compounds? what is the meaning of red in highlight?
4. Figure numbers are not matched with the manuscript.
5. mM means 0.001 mmol/ml, be careful in the concentration of H2O2, and WTA in Figs
6. cleave caspase 3-> cleaved caspase 3
7. All blot images should have size makers or provide entire uncropped images as supplementary figures.
8. Fig. 2: did not indicate multiple comparisons between groups.
9. Table 1 has no legend. Also, it can be consolidated with Fig. 4D.
10. Fig. 3A: Scrambled or vehicle? Need to compare H2O2 ± z-VAD-FMK and WFA.
11. Fig. 3A right panel: please check the y-axis. Only 0.3% of apoptosis after treatment of H2O2?
12. Fig. 3B: immunofluorescence data has no legend, DIPA -> DAPI.
13. Fig. 5A: HO-1 blot is not matched with the bar graph on the right panel, decreased HO-1 in WFA-treated
14. Fig. 5B-C: keep consistent either AKT or Akt, vehicle or scrambled.
15. Fig. 6C: it is important to interpret whether the inhibition of pAKT by MK2206 exacerbates cell apoptosis. Please compare the groups between H2O2 in the absence or presence of MK2206.
Author Response
Point-to-point Responses to Reviewer #2
Comments and Suggestions for Authors
The study investigated the effect of withaferin A, an active compound isolated from plants, in oxidative stress-mediated death of HRMECs. The authors investigated its antioxidant effect using hypoxia and hydrogen peroxide in vitro as well as a mouse model of ischemia/reperfusion injury. Withaferin A protects against cell apoptosis by increasing HO-1 and Prdx-1 expression via AKT phosphorylation. The in vitro mechanism study is solid, and the finding is interesting. However, in vivo data could not provide any evidence for the therapeutic possibilities against retinal endothelial cell protection since they used whole retinal homogenates. The author did not assess how much withaferin A reaches to the retina after systemic injection. (e.g. What is the rationale of 10 mmol/g bw?). Whole retinal flat mount immunostaining can be a useful tool to evaluate the apoptosis of retinal endothelium. Furthermore, the manuscript is not fully ready to publish. The authors are encouraged to edit the manuscript carefully. E.g. revise enormous errors in a text such as abbreviations, the concentration of chemicals, statistical analysis, and lack of information in the legends.
Response: We appreciated the reviewer’s positive comments and the constructive suggestions which significantly help us to strengthen the study.
We agree with the reviewer that whole retinal flat mount immunostaining is a useful tool to evaluate the apoptosis of retinal endothelium. However, we had met a difficulty in doing this experiment. The possible reasons are as follows: 1. Retina has inner limiting membrane barriers so blood vessels in retina could not be stained easily. 2. Due to the complex three-dimensional structure of the retina consisting of 10 layers and multilayered blood vessels, it is hard to accurately visualize the apoptosis of retinal endothelial cells in the whole layers of the retina. Thus, we used whole retinal homogenates for preliminary verification in vivo. This way has also been used by other researchers(1, 2). We mentioned this point in the Discussion and cited the reference.
We are sorry that we did not perform the pharmacokinetic experiments of withaferin A, and therefore did not know how much withaferin A reached the retina after systemic injection. This study focused on the qualitative effect and the underlying mechanism of withaferin A. The dose of 10 mmol/g BW was based on the calculation and conversion from the in vitro concentration of withaferin A. We appreciate the reviewer's constructive suggestions and the pharmacokinetic study of withaferin A may be performed in an the subsequent study.
There are representative concerns:
Q1. Methods: missing catalog numbers, city, and country in some reagents, assay kits, secondary antibodies, and primer sequences.
A1. Thank you for reminding us for this point. They have been added in the revised version. Primer sequences are provided in the Supplementary Material.
Q2. Method 2.9: note animal numbers, anesthesia, ARVO Statement for the Use of Animals in Ophthalmic and Vision Research, IACUC protocol number
A2. Thank you for reminding us for this point. These items have been added in the revised version.
Q3. Fig. 1: did not interpret what does radar graph means, any positive control compounds? what is the meaning of red in highlight?
A3. We are sorry that our original description on the radar graph of Figure 1 is too terse. The bioavailability radar allows for the preview of the drug-likeness of the molecules of interest. The six axes of Bioavailability Radar represent six important properties for bioavailability. LIPO (lipophilicity), SIZE, POLAR (polarity), INSOLU (insolubility), INSATU (insaturation), and FLEX (flexibility) are the six physicochemical characteristics that are taken into account. Each property is defined by a descriptor of SwissADME and a range of optimal values highlight in pink. The red line of the compound under study must be fully included in the pink area, any deviation represents a suboptimal physicochemical property for oral bioavailability. The description has been added in the methods part of manuscript.
Q4. Figure numbers are not matched with the manuscript.
A4. We thank the reviewer for noting the error of the figure numbers, which has been corrected in the revised version.
Q5. mM means 0.001 mmol/ml, be careful in the concentration of H2O2, and WTA in Figs
A5. We thank the reviewer for noting the error, which has been corrected in the R1 version.
Q6. cleave caspase 3-> cleaved caspase 3
A6. Thank you for noting the error, which has been corrected in the revised version.
Q7. All blot images should have size makers or provide entire uncropped images as supplementary figures.
A7. When we submitted the manuscript, the entire uncropped images has been uploaded in the initial submission following the author’s instructions. If the reviewer could not find them, we can submit them again.
Q8. fig. 2: did not indicate multiple comparisons between groups.
A8. As suggested, multiple comparisons between groups have been indicated in fig. 2.
Q9. Table 1 has no legend. Also, it can be consolidated with Fig. 4D.
A9. We thank the reviewer for these insightful comments. Table 1 has been consolidated with figure Fig. 4, and the legend has been added.
Q10. Fig. 3A: Scrambled or vehicle? Need to compare H2O2 ± z-VAD-FMK and WFA.
A10. We thank the reviewer for careful reading, it is vehicle, which has been corrected. Meanwhile, H2O2 ± z-VAD-FMK and WFA has been compared.
Q11. Fig. 3A right panel: please check the y-axis. Only 0.3% of apoptosis after treatment of H2O2?
A11. We thank the reviewer for noting the error, it should be 30%, not 0.3%, which has been corrected.
Q12. Fig. 3B: immunofluorescence data has no legend, DIPA -> DAPI.
A12. Thank you for reminding us for this point. The legend has been added in the revised version.
Q13. Fig. 5A: HO-1 blot is not matched with the bar graph on the right panel, decreased HO-1 in WFA-treated
A13. Sorry for this error, I rotated the picture by mistake when I finally assembled the picture, which has been corrected.
Q14. Fig. 5B-C: keep consistent either AKT or Akt, vehicle or scrambled.
A14. We thank for the reviewers careful reading, which has been corrected.
Q15. Fig. 6C: it is important to interpret whether the inhibition of pAKT by MK2206 exacerbates cell apoptosis. Please compare the groups between H2O2 in the absence or presence of MK2206.
A15. We greatly appreciate the reviewer's constructive suggestions. As requested, H2O2 in the absence or presence of MK2206 has been compared and results are shown in Fig. 6D in the revised version.
- Santana-Garrido Á, Reyes-Goya C, Fernández-Bobadilla C, Blanca AJ, André H, Mate A, Vázquez CM. NADPH oxidase–induced oxidative stress in the eyes of hypertensive rats. Molecular Vision. 2021;27:161.
- Hsu Y-J, Lin C-W, Cho S-L, Yang W-S, Yang C-M, Yang C-H. Protective Effect of Fenofibrate on Oxidative Stress-Induced Apoptosis in Retinal–Choroidal Vascular Endothelial Cells: Implication for Diabetic Retinopathy Treatment. Antioxidants. 2020;9(8):712.
Round 2
Reviewer 1 Report
Dear Authors,
I am very pleased to see that you have appreciated our suggestions.
Now it's very good paper. Congratulations for the work done with great precision and innovation.
Kind regards
Author Response
Response: We are pleased to the reviewer’s appreciation on our manuscript merit.

Reviewer 2 Report
The authors have responded adequately to the concerns raised by this reviewer. There remain only a few points that still need attention.
1. Please check the concentration of hydrogen peroxide the author used. The millimolar concentration of H2O2 is too harsh for cells. If this is true, please provide any references or rationale.
2. Line 145: Cat#9803
3. Cell Signaling Technology
4. Line 214: nmoles/g
5. The legends still don't provide sufficient information.
6. Results 3.4: Please carefully note the concentration of inhibitors. 10 M -> 10 μM
7. Fig. 7. MK2206 inhibited pAKT levels in the retina tissue homogenates? Furthermore, Fig. 7B-C showed that even though WFA blocked I/R-mediated nitrotyrosine expression, it marginally decreased caspase-3 activity. The authors have to address this in the results and discussion.
8. Line 482: Ref (29) is superscripted.
Author Response
Rebuttal Letter
Point-to-point Responses to Reviewer #2
Comments and Suggestions for Authors
The authors have responded adequately to the concerns raised by this reviewer. There remain only a few points that still need attention.
Response: We greatly appreciate this reviewer’s positive assessment on our manuscript and agree with all his/her expert suggestions. Specifically, we have made the following revisions:
Q1. Please check the concentration of hydrogen peroxide the author used. The millimolar concentration of H2O2 is too harsh for cells. If this is true, please provide any references or rationale.
A1. We thank the reviewer for noting the error of the concentration, which should be µmol/l, not µmol/ml, and it has been corrected in the R2 version.
Q2. Line 145: Cat#9803
A2. We thank the reviewer for noting the error, which has been corrected in the R2 version.
Q3. Cell Signaling Technology
A3. As suggested, it has been corrected in the R2 version.
Q4. Line 214: nmoles/g
A4. As suggested, it has been corrected in the R2 version.
Q5. The legends still don't provide sufficient information.
A5. Thank you for reminding us for this point. The detailed legends have been added in figure3B and figure4D.
Q6. Results 3.4: Please carefully note the concentration of inhibitors. 10 M -> 10 μM
A6. We thank the reviewer for noting the error, which has been corrected in the R2 version.
Q7.Fig. 7. MK2206 inhibited pAKT levels in the retina tissue homogenates? Furthermore, Fig. 7B-C showed that even though WFA blocked I/R-mediated nitrotyrosine expression, it marginally decreased caspase-3 activity. The authors have to address this in the results and discussion.
A7. The reviewer is absolutely right, MK-2206 inhibited phosphorylation of Akt S473, as well as prevented Akt-mediated phosphorylation of down-stream signaling molecules (1). We appreciate the reviewer’s suggestion, and added the indicated information in the results and discussion.
(1)Yan L. Abstract# DDT01-1: MK-2206: A potent oral allosteric AKT inhibitor[J]. Cancer research, 2009, 69(9_Supplement): DDT01-1-DDT01-1.
Q8. Line 482: Ref (29) is superscripted.
A8. As suggested, it has been corrected.
